# A Serological Analysis of the Humoral Immune Responses of Anti-RBD IgG, Anti-S1 IgG, and Anti-S2 IgG Levels Correlated to Anti-N IgG Positivity and Negativity in Sicilian Healthcare Workers (HCWs) with Third Doses of the mRNA-Based SARS-CoV-2 Vaccine: A Retrospective Cohort Study

**DOI:** 10.3390/vaccines11071136

**Published:** 2023-06-23

**Authors:** Nicola Serra, Maria Andriolo, Ignazio Butera, Giovanni Mazzola, Consolato Maria Sergi, Teresa Maria Assunta Fasciana, Anna Giammanco, Maria Chiara Gagliano, Antonio Cascio, Paola Di Carlo

**Affiliations:** 1Department of Public Health, University Federico II of Naples, 80131 Napoli, Italy; 2Clinical Pathology Laboratory, Provincial Health Authority of Caltanissetta, 93100 Caltanissetta, Italy; 3Degree Course in Medicine and Surgery, Medical Scholl of Hypatia, University of Palermo, 93100 Caltanissetta, Italy; 4Infectious Disease Unit, Provincial Health Authority of Caltanissetta, 93100 Caltanissetta, Italy; 5Department of Pathology and Laboratory Medicine, University of Ottawa, 401 Smyth Road, Ottawa, ON K1H 8L1, Canada; 6Department of Health Promotion, Maternal-Childhood, Internal Medicine of Excellence “G. D’Alessandro”, University of Palermo, 90127 Palermo, Italy; 7Infectious Disease Unit, Department of Health Promotion, Maternal-Childhood, Internal Medicine of Excellence “G. D’Alessandro”, University of Palermo, 90127 Palermo, Italy

**Keywords:** healthcare workers, SARS-CoV-2, COVID-19, anti-S2 IgG, anti-S1 IgG, anti-RBD IgG and anti-N IgG, hybrid immunity

## Abstract

Background: With SARS-CoV-2 antibody tests on the market, healthcare providers must be confident that they can use the results to provide actionable information to understand the characteristics and dynamics of the humoral response and antibodies (abs) in SARS-CoV-2-vaccinated patients. In this way, the study of the antibody responses of healthcare workers (HCWs), a population that is immunocompetent, adherent to vaccination, and continuously exposed to different virus variants, can help us understand immune protection and determine vaccine design goals. Methods: We retrospectively evaluated antibody responses via multiplex assays in a sample of 538 asymptomatic HCWs with a documented complete vaccination cycle of 3 doses of mRNA vaccination and no previous history of infection. Our sample was composed of 49.44% males and 50.56% females, with an age ranging from 21 to 71 years, and a mean age of 46.73 years. All of the HCWs’ sera were collected from April to July 2022 at the Sant’Elia Hospital of Caltanissetta to investigate the immunologic responses against anti-RBD, anti-S1, anti-S2, and anti-N IgG abs. Results: A significant difference in age between HCWs who were positive and negative for anti-N IgG was observed. For anti-S2 IgG, a significant difference between HCWs who were negative and positive compared to anti-N IgG was observed only for positive HCWs, with values including 10 (U/mL)–100 (U/mL); meanwhile, for anti-RBD IgG and anti-S1 IgG levels, there was only a significant difference observed for positive HCWs with diluted titers. For the negative values of anti-N IgG, among the titer dilution levels of anti-RBD, anti-S1, and anti-S2 IgG, the anti-S2 IgG levels were significantly lower than the anti-RBD and anti-S1 levels; in addition, the anti-S1 IgG levels were significantly lower than the anti-RBD IgG levels. For the anti-N IgG positive levels, only the anti-S2 IgG levels were significantly lower than the anti-RBD IgG and anti-S1 IgG levels. Finally, a logistic regression analysis showed that age and anti-S2 IgG were negative and positive predictors of anti-N IgG levels, respectively. The analysis between the vaccine type and mixed mRNA combination showed higher levels of antibodies in mixed vaccinated HCWs. This finding disappeared in the anti-N positive group. Conclusions: Most anti-N positive HCWs showed antibodies against the S2 domain and were young subjects. Therefore, the authors suggest that including the anti-SARS-CoV-2-S2 in antibody profiles can serve as a complementary testing approach to qRT-PCR for the early identification of asymptomatic infections in order to reduce the impact of potential new SARS-CoV-2 variants. Our serological investigation on the type of mRNA vaccine and mixed mRNA vaccines shows that future investigations on the serological responses in vaccinated asymptomatic patients exposed to previous infection or reinfection are warranted for updated vaccine boosters.

## 1. Introduction

From the beginning of the SARS-CoV-2 pandemic to the last Omicron BA.4 and BA.5 epidemic waves, the characteristics and handling of the pandemic have evolved. Several factors, including increasing immunity within the population and the availability of effective vaccines, rapid tests, and treatments, have contributed to the fight against the spread of Omicron subvariants [1,2]. Approximately 49 million subjects underwent the primary vaccination cycle or the three-dose vaccination schedule with the Comirnaty vaccine and the Spikevax mRNA COVID-19 vaccine. These vaccines are available on different mRNA vaccine platforms and were developed by Pfizer/BioNTech and Moderna [1,3,4].

Regardless of their age and underlying pathological conditions, healthcare workers (HCWs) have been identified as a group of interest [5,6,7]. The current SARS-CoV-2 pandemic history has highlighted the crucial role of serological investigations to assess variations in the humoral response. This allows researchers to study how the humoral response varies according to age, gender, and underlying disease in HCWs [8,9,10,11]. With the advent of the vaccination campaign, including an updated booster version of the vaccine for Omicron in the autumn of 2022, the usefulness of the serological survey has evolved, which allows researchers to differentiate natural immunization from immunization due to vaccination by using the seropositivity to N-antigen approach [10,11]. Vaccination and natural infection generate serological antibody responses that vary widely [11,12,13]. Many epidemiological studies on healthcare workers have been conducted to evaluate the levels of antibodies against some structural protein components of the virus after vaccination and natural infection [10,11,14,15].

The combination of the immune response to natural infection and that induced by vaccination, called hybrid immunity, has recently become a subject of study and discussion by researchers [16]. Healthcare workers proved to be an ideal population in which to study the hybrid immune response, as they are continuously exposed to virus variants and are vaccinated in accordance with vaccination scheme programs that are performed at adequate and standardized times [16]. Recently, different serology tests have become available to identify neutralizing SARS-CoV-2-specific antibodies [17]. Thus, the literature shows significant variability in the titers of antibodies against the anti-SARS-CoV-2 anti-nucleocapsid (N), anti-receptor-binding domain (RBD), and the anti-SARS-CoV-2 spike protein S1 and S2 subunit [11,14,18].

The availability of multiplex antibody serology against all four antigen targets can provide a composite SARS-CoV-2 IgG antibody result with high specificity. This can benefit complex clinical presentations, epidemiological research, and decisions related to infection prevention strategies [11,19,20,21].

### Objective

This study investigated the antibody levels for spike 1 (S1), spike 2 (S2), and the receptor-binding domain (RBD) SARS-CoV-2 antigens in healthcare workers with no previous history of infection after six months following their third dose of the SARS-CoV-2 mRNA vaccine (Pfizer/Biontech, Moderna) or mixed mRNA vaccine. Moreover, we looked at the antibody responses to nucleocapsids to control for possible confounding infections with SARS-CoV-2 and the antibody profiles according to the vaccine type and mixed mRNA vaccine combination.

## 2. Materials and Methods

### 2.1. Patients and Study Design

The antibody response was retrospectively evaluated via multiplex assays in a sample of 538 asymptomatic healthcare workers (HCWs) at the Sant’Elia Hospital of Caltanissetta (Caltanissetta, Italy), a referral hospital of the Caltanissetta area for antibody screening of HWCs. The HWCs had been vaccinated with the primary cycle (three doses of mRNA SARS-CoV-2 vaccination) and had no known history of SARS-CoV-2 infection. The information on SARS-CoV-2 infection was collected through a survey that asked the sample subjects to indicate the date of their last positive COVID-19 swab nasopharyngeal test, the last mRNA vaccine jab, and the hospital department where they worked. Any positive case that was identified through molecular PCR and rapid antigen tests (swabs) was excluded.

All of the HCWs had been tested for their immunologic responses against SARS-CoV-2 infection from serum samples taken six months after the third dose of the SARS-CoV-2 vaccine, from April to July 2022. These HCWs comprised 49.44% males and 50.56% females, with ages in the range of 21 to 71 years and a mean age of 46.73 years.

Following the Italian vaccination program, the HCWs received three doses of the mRNA vaccine (Comirnaty (BNT162b2) and Spikevax). In particular, 332 HCWs received the Pfizer/BionTech vaccine only, 88 HCWs received the Moderna vaccine only, and 44 HCWs received a mixed mRNA vaccine combination (Pfizer/BionTech and Moderna).

The HCW sample was stratified according to the positivity for N-abs to differentiate those who had hybrid immunity from those who had only vaccinal immunity.

The group with hybrid immunity that tested positive for SARS-CoV-2 infection (COVID-19 H) was composed of 186 subjects (34.57%) with an anti-nucleocapsid (N) protein IgG level ≥10 U/mL, which included 44.62% males and 55.38% females, with ages ranging from 23 to 67 years (mean of 43.88 and standard deviation equal to 12.10 years).The group with vaccine immunity that tested negative for SARS-CoV-2 infection (COVID-19 V) was composed of 352 subjects (65.43%) with an anti-nucleocapsid (N) protein IgG level <10 U/mL, which included 51.99% males and 48.01% females, with ages ranging from 23 to 73 years (mean of 48.24 and standard deviation equal to 12.17 years).

### 2.2. Titration of SARS-CoV-2 Infection Antibody Analysis

The qualitative detection (IgG screening) and semi-quantitative (U/mL) detection of IgG class antibodies against SARS-CoV-2 were undertaken using the BioPlex 2200 SARS-CoV-2 IgG Panel (Bio-Rad Laboratories, Inc., Hercules, CA, USA). This device can screen and differentiate IgG antibodies to the receptor-binding domain (RBD) and the spike 1 (S1), spike 2 (S2), and nucleocapsid proteins (N) of the SARS-CoV-2 coronavirus [19,20]. The positive anti-SARS-CoV-2 N/RBD/S1/S2 IgG levels were set to ≥10 U/mL, particularly for levels that were above 100 U/mL, and the machine provided 1:16 dilutions of the anti-IgG levels. The machine’s upper limit of values obtained from the dilutions was set at 1600 U/mL. All of the calculations necessary to interpret the results were performed automatically by Plex^TM^ 2200 System Software (Bio-Rad, USA) [19,20,21].

### 2.3. Statistical Analysis

The data are presented as numbers or percentages for categorical variables. The continuous data are expressed as the mean ± standard deviation (SD), or the median with interquartile range (IQR). The chi-square test and Fisher’s exact test were performed to evaluate significant differences in the proportions or percentages between the two groups. Fisher’s exact test was used where the chi-square test was not appropriate. Multiple comparison Cochran’s Q tests were used to compare the differences among percentages for paired data, considering the null hypothesis that there are no differences between the variables or modalities. When Cochran’s Q test was positive (*p*-value < 0.05), a minimum required difference for a significant difference between the two proportions was calculated using the minimum required differences method with the Bonferroni *p*-value corrected for multiple comparisons. The test for a normal distribution was performed via the Shapiro–Wilk test. The Kruskal–Wallis test followed by the post hoc test with the Conover test for pairwise comparisons were performed in multi-comparisons among three or more independent samples in the case that no normal distribution was found. Particularly, where the tests on medians showed a significant difference and the medians were equal, the mean rank values were described.

Logistic regression analysis was used to find the best-fitting model to describe the relationship between the dichotomy-dependent variable and two or more variables. In addition, the possible associations between two non-normal variables, such as anti-N IgG (dependent variable) and predictors such as age, anti-RBD IgG, anti-S1 IgG, and anti-S2 IgG levels were evaluated. Particularly, for this step, we defined the following variables:Anti-N IgG (dependent variable): anti-N IgG level < 10 U/mL = 0 (negative) and anti-N IgG level ≥ 10 U/mL = 1 (positive);Anti-RBD IgG: anti-RBD IgG level < 10 U/mL = 0 and anti-RBD IgG level ≥ 10 U/mL = 1;Anti-S1 IgG: anti-S1 IgG level < 10 U/mL = 0 and anti-S1 IgG level ≥ 10 U/mL = 1;Anti-S2 IgG: anti-S2 IgG level < 10 U/mL = 0 and anti-S2 IgG level ≥ 10 U/mL = 1.

All tests with a *p*-value < 0.05 were considered significant. The statistical analyses were carried out with Matlab statistical toolbox version 2008 (MathWorks, Natick, MA, USA) for 32-bit Windows.

## 3. Results

Table 1 shows the clinical information of our sample, including the age, gender, antibody pattern, and anti-SARS-CoV-2 IgG levels in the vaccinated healthcare worker subjects. Table 1 shows the general characteristics of 538 vaccinated healthcare workers and the detection of antibodies against the receptor-binding domain (RBD); the detection of antibodies against the S protein, S1 domain, and S2 domain; and the detection of the level of antibodies against the nucleocapsid (N) protein of SARS-CoV-2.

In our sample, the mean and median ages of the HCWs were greater than 45 years, and no difference in gender was observed. According to the definition of a positive response against the nucleocapsid (N) protein of SARS-CoV-2 IgG ≥ 10 U/mL, 186/538 (34.6%) of the enrolled subjects had a positive anti-N response. In particular, 167/538 (31%) had an anti-N in the range of 10–100, and 19/538 (3.5%) showed a titer >100 U/mL of IgG anti-SARS-CoV-2 N. The detection of antibodies against the receptor-binding domain (RBD) was >100 U/mL in 530/538 (98.5%) of the HCWs, and only in two subjects (0.4%) was the response <10 U/mL. Similarly, for the detection of antibodies against the S protein, the S1 protein domain was >100 U/mL in 537/538 (99.8%) of the vaccinated enrolled HCWs. The detection of antibodies against the S2 protein domain was ≥10 U/mL in 70% of the HCWs, and the titer was >100 U/mL in 87/538 of the subjects (16.2%).

Table 2 shows our sample of HCWs considering two groups: COVID-19-H, including all HCWs with hybrid immunity, who showed an anti-N IgG level ≥10 U/mL, while the COVID-19-V group, including HCWs with vaccine immunity only, showed an anti-N IgG level <10 U/mL. For both groups, we reported parameters such as age, gender, and the mean and median levels of the anti-SARS-CoV-2 anti-nucleocapsid (N) protein, considering that all measures were stratified into three intervals based on a score <10 U/mL (negative), 10 U/mL and 100 U/mL (positive), and >100 U/ mL (positive with diluted levels). This classification was performed to reduce errors by considering both the diluted and undiluted positive values. It was possible to compute the actual values from the diluted values, but the machine had an upper limit of 1600 U/mL, and this limit could introduce statistical bias into the results.

In Table 2, we observed a significant difference in age between negative and positive HCWs to anti-N IgG (median: 48 years vs. 43 years, *p* = 0.0001).

By analyzing the negative (63 N, 289 P) and positive anti-S2 IgG (9 N, 176 P) cases in the groups with negative and positive anti-N IgG, a significant relationship was observed (*p* < 0.0001).

Comparing the negative and positive cases for anti-N IgG, a significant difference for the anti-RBD IgG-diluted titer (mean rank: 246.1 vs. 301.7, *p* < 0.0001) and the anti-S1 IgG-diluted titer was observed (mean rank: 232.4 vs. 338.6, *p* < 0.0001).

Finally, for anti-S2 IgG, a significant difference between the negative and positive cases for anti-N IgG was observed only for positive HCWs, with levels in the 10 (U/mL)–100 (U/mL) range (median: 28 vs. 47, *p* < 0.0001).

As shown in Table 3, we investigated the significant parameters in Table 2 to individualize by logistic regression the significant predictors of anti-N IgG.

The logistic regression analysis showed that age and anti-S2 IgG were the negative and positive significant predictors of anti-N IgG, respectively. In other words, young HCWs had anti-N IgG levels ≥10 U/mL (OR: 0.97, *p* = 0.0001), anti-S2 IgG levels ≥10 U/mL were associated with anti-N IgG levels ≥10 U/mL, and anti-S2 IgG levels <10 U/mL were associated with anti-N IgG levels <10 U/mL (OR:4.5, *p* = 0.0001).

In Table 4, we reported the percentages of positive patients for anti-N IgG, anti-RBD IgG, anti-S1 IgG, and anti-S2 IgG.

In Table 4, we found a lower percentage of healthcare workers who were positive for anti-N IgG compared to anti-RBD IgG, anti-S1 IgG, and anti-S2 IgG, (34.6% vs. 99.6%, 34.6% vs. 99.8%, and 34.6% vs. 86.6%, respectively). Similarly, we found a lower percentage of healthcare workers who were positive for anti-S2 IgG levels compared to anti-RBD IgG and anti-S1 IgG levels (86.6% vs. 99.6%, 86.6% vs. 99.8%, respectively).

In Table 5, we reported the diluted levels of anti-RBD IgG, anti-S1 IgG, and anti-S2 IgG for positive and negative levels of anti-N IgG.

In Table 5, we observed for the anti-N IgG negative levels that among the titer dilution levels for anti-RBD IgG, anti-S1 IgG, and anti-S2 IgG, the levels of anti-S2 IgG were significantly lower than the anti-RBD IgG and anti-S1 IgG levels (178.2 vs. 1432.6 and 178.2 vs. 1242.3, respectively), and the anti-S1 IgG levels were significantly lower than the anti-RBD IgG levels (1242.3 vs. 1432.6). Instead, for the anti-N IgG positive levels, only the levels of anti-S2 IgG were significantly lower than the anti-RBD IgG and anti-S1 IgG levels (132.9 vs. 1582.2 and 132.9 vs. 1575.7, respectively).

The results of the anti-RBD IgG, anti-S1 IgG, and anti-S2 IgG levels associated with three doses of either mRNA cycle vaccination type and the mixed mRNA combination are shown in Figure 1. Regarding the anti-RBD IgG levels, a significant difference among the Pfizer/BionTech, Moderna cycle vaccination and the mixed mRNA combination was observed (*p* = 0.00014). In particular, the post hoc Kruskal–Wallis test showed less significant levels of anti-RBD IgG for the Pfizer/BionTech cycle vaccination in comparison to the Moderna cycle vaccination and mixed mRNA combination (mean rank: 255.1 vs. 286.7, *p* < 0.05 and 255.1 vs. 297.1, *p* < 0.05, respectively). For the anti-S1 IgG levels, we found higher levels in HCWs vaccinated with the mixed mRNA combination than in those who received the Pfizer/BionTech and Moderna cycle vaccinations (mean rank: 314.1 vs. 251.5, *p* < 0.05; 314.1 vs. 277.8, *p* < 0.05, respectively). HCWs vaccinated with the mixed mRNA combination showed anti-S2 IgG levels that were greater than those vaccinated with single Moderna and Pfizer/BionTech (mean rank: 320.2 vs. 291.1, *p* < 0.05; 320.2 vs. 245.8, *p* < 0.05, respectively).

In Figure 2 and Figure 3, the anti-RBD IgG, anti-S1 IgG, and anti-S2 IgG levels associated with vaccine type taken by healthcare workers (HCWs) who were positive and negative for the anti-N (nucleocapsid) protein after the third dose of a COVID-19 vaccination are shown.

Figure 2 shows that for the anti-RBD IgG levels, there was a significant difference among the Pfizer/BionTech, Moderna, and mixed mRNA combination (*p* = 0.00018). In particular, the post hoc Kruskal–Wallis test showed a lower anti-RBD IgG level in the Pfizer/BionTech cycle vaccination in comparison to the Moderna and mixed mRNA combination (mean rank: 163.8 vs. 192.5, *p* < 0.05 and 163.8 vs. 203.1, *p* < 0.05, respectively). For the anti-S1 IgG levels, higher levels were found in HCWs with the mixed mRNA combination in comparison to those who received the Pfizer/BionTech and Moderna cycle vaccinations (mean rank: 223.3 vs. 159.7, *p* < 0.05; 223.3 vs. 182.1, *p* < 0.05, respectively). For the anti-S2 IgG levels, lower levels were found in HCWs vaccinated with the Pfizer/BionTech in comparison to the Moderna and mixed mRNA combination (mean rank: 160.1 vs. 191.2, *p* < 0.05; 160.1 vs. 215.7, *p* < 0.05, respectively).

Figure 3 shows that for the anti-RBD IgG, anti-S1 IgG, and anti-S2 IgG levels, there were no significant differences among the Pfizer/BionTech, Moderna, and mixed mRNA combination (anti-RBD IgG mean rank: 94, 93.3, 93.9, *p* = 0.98, respectively; anti-S1 IgG mean rank: 94.3, 94, 91.2, *p* = 0.74, respectively; anti-S2 IgG mean rank: 86.6, 98.2, 106.9, *p* = 0.09, respectively).

## 4. Discussion

Utilizing a new multiplexed antibody detection approach in this study, we observed the induction of diverse anti-spike antibodies after the third dose of spike protein-encoding Moderna and Pfizer/BioNTech vaccines, including a response to nucleocapsids, and provided actionable information about the characteristics of specific humoral immunity against SARS-CoV-2 in HCWs who experienced asymptomatic infection.

Hybrid immunity provides better protection against subsequent SARS-CoV-2 variant infections than either vaccination or infection alone [16,22]. However, the antibody profile picture is complex as a result of the population’s multifaceted immunity pattern due to large-scale vaccination and infections with viral variants.

Omicron variants have been reported to have potent immune evasion against vaccine-induced neutralizing antibodies, and increasing evidence supports the crucial role of the T cell response to SARS-CoV-2 in controlling the disease [23]. According to previous research reports, the Omicron variant has a strong ability to escape humoral immunity, especially in patients with dysfunctional human immune responses [24]. Recent developments in immunotherapy in viral infections, such as adoptive cell transfer (ACT) with chimeric antigen receptor (CAR) T cells, could represent a prominent example [25] for the treatment of COVID-19.

In this way, the study of the humoral responses of healthcare workers, a population that is immunocompetent, adherent to vaccination, and continuously exposed to the different virus variants, can help us understand immune protection and establish goals in vaccine design [8].

Although we analyzed HCWs with no medical history or positive molecular tests for SARS-CoV-2 infection, we found a significant titer against anti-N in 34.6% of the enrolled HCWs after the third dose of an mRNA SARS-CoV-2 vaccination. Of these, 3.5% showed a higher titer (>100 U/mL).

This confirms that the transmission of SARS-CoV-2 after a third dose of vaccines among populations without underlying diseases causes asymptomatic infections. Receiving three doses of an mRNA vaccine, relative to being unvaccinated and receiving two doses, was associated with protection against both the Omicron and Delta variants [1,3,26,27,28].

Although, how long N-antibodies persist after infection is unknown; they appear to decrease very slowly. Further long follow-up studies are needed [29].

Neutralizing antibodies (NAbs) target the SARS-CoV-2 S protein and/or the RBD to neutralize viral binding to ACE2 receptors of potential host cells [12,13,14,26]. Regarding the antibody response against the spike protein receptor-binding domain (RBD), all of the HCWs except two produced anti-RBD antibodies, and 98.5% had a higher titer, confirming that the vaccines elicited more antibodies against the spike protein receptor-binding domain (RBD). Moreover, all of the HCWs enrolled in our study except one showed a higher titer of the anti-S1 protein, and the levels of anti-S1 were higher than the anti-RBD titer. These findings confirm an anti-RBD and anti-S1 response after vaccination with either the Pfizer/BioNTech or Moderna vaccine [11,21,26,28,30,31].

Comparing the patients’ positive anti-N versus negative anti-N levels, we found the anti-RBD and anti-S1 titers to be significantly higher in anti-N-positive HCWs. This is in accordance with other studies showing that hybrid immunity may be significantly associated with higher antibody titers [22,26,32,33]. This may be consistent with the maturation of spike-specific antibodies that have been reported after SARS-CoV-2 infection [34,35,36].

Regarding the anti-S2 domain, we found that 13.5% of the enrolled subjects had undetectable titers, while 70% showed a positive titer in the 10–100 U/mL range and 16.2% > 100 U/mL.

We found few studies that compared the prevalence of N-Abs, RBD, and S2 antibodies, especially in HCWs [14,37]. Błaszczuk et al. reported that anti-N and anti-S2 antibodies were lower than anti-RBD titers in HCWs who were vaccinated with two doses of the Pfizer vaccine [14].

When we compared the humoral response to the spike S1, S2, and nucleocapsid proteins after SARS-CoV-2 infection in HCWs, we found a higher titer for the anti-S2 IgG domain in patients who were positive for the anti-N IgG response within subjects with titers that were within 10–100 U/mL. Namely, vaccinated HCWs with an immune history of SARS-CoV-2 infection (anti-N positive group) showed a higher percentage of positive tests only for the anti-S2 domain (Table 2).

We observed from a logistic regression standpoint that age and anti-S2 IgG were negative and positive significant predictors of anti-N IgG, respectively (Table 3). In particular, the positive anti-S2 subjects, regardless of their titer values or antibody levels, were associated with positive anti-N and, therefore, infection.

The anti-S IgG assay displayed good sensitivity and specificity in discriminating subjects with breakthrough infections (BIs), especially in the recent period; however, the anti-S IgG assay may have a low PPV in this HCW setting [5,10]. In this way, the anti-S2 and anti-N IgG associations we found in vaccinated HCWs may help discriminate subjects with early infection. This finding is in accordance with Liao et al. [38], who reported that detecting IgG against the S2 protein could supplement nucleic acid testing to identify asymptomatic patients. Therefore, the authors suggest including anti-S2 IgG in antibody profiles in HCWs for the early identification of asymptomatic infections, which may reduce the impact of the future SARS-CoV-2 variant pandemic in the hospitalized frail population. The higher sensitivity of S2-mAbs (IgG) compared to RBD-mAbs (IgG) could supplement nucleic acid testing in identifying asymptomatic cases at an early period and in frail populations, as suggested in another study [39].

The S2 subunit of the SARS-CoV-2 spike protein has aroused much interest for its role in S2 as a potential immunogen in studying antibodies and cell-mediated responses [40,41,42,43].

The high percentage of subjects who were positive for the S2 domain in patients with the anti-N protein is due to the structural and functional role of the S2 domain. Compared to the S1 subunit, the membrane-anchored S2 subunit, which mediates viral and cell membrane fusion through receptor-induced conformational rearrangements, shows higher protein sequence conservation among the COVID-19 spike proteins. Therefore, the anti-S2 IgG response is always present, regardless of vaccination and the virus variants that cause infection [42,43,44].

Concerning the age variable in our study, younger healthcare workers showed better hybrid immunity than older subjects. Young HCWs are often asymptomatic, as their lifestyle exposes them to more contact with subjects with SARS-CoV-2 [45,46,47].

Apart from untreatable factors (age, sex, race), other risk factors that underlie chronic disease, such as an improper diet, tobacco consumption, excessive alcohol consumption, insufficient physical activity, sedentary behavior, and personal or professional stress, impact SARS-CoV-2 infections among HCWs, and should be analyzed during the COVID-19 pandemic period [48]. In our study, the survey information on SARS-CoV-2 infection included basic demographic characteristics, with only age and sex analyzed in our statistical database.

According to Table 4, we found a high percentage of antibody response against RBD and S1, while the percentage response against the S2 domain was lower. Our results are in accordance with those observed in fully vaccinated individuals who demonstrated an average of 50-fold higher antibody levels than naturally infected unvaccinated individuals, with immunity reacting strongly to RBD/S1 and weakly to S2 [11,21,49,50].

The RBD, S1, and S2 antibodies in high titer dilutions in HCWs with hybrid immunity versus vaccinated HCWs showed low levels of anti-S2 in the negative and positive anti-N groups (Table 5). As discussed above, this confirms that different immunoassays result as the antibodies of the S2 subunit are less frequently elicited. A recent investigation of the S IgG level in SARS-CoV-2 patients showed high reactivity for S1, while no antibodies were detected against the heptad repeat domain 2 of S2 [50]. The difficulty of interpreting the dynamic response to anti-S2 [18] is in tandem with the efforts of the scientific community to explore the potential benefits of anti-S2 protection for extended protection versus future Omicron variants [51,52].

During waves of the pandemic, the emergence of new variants that were partially resistant to available vaccines and the report of adverse reactions forced developing and industrialized countries to start a mix of COVID-19 vaccines, with the hope of immunizing a greater percentage of people; thus, a mix of mRNA COVID-19 vaccines became involved. In our study, the analysis between the vaccine type and a mixed mRNA vaccine combination on anti-RBD IgG, anti-S1 IgG, and anti-S2 IgG levels confirmed in all of the enrolled HCWs that a mixed mRNA combination provided greater protection than the use of a single type of vaccine. The authors consider this to be of great interest because, in pandemic emergencies, the possibility of performing a vaccination cycle is determined by the local availability of the vaccine platform. The underlying mechanism for higher immunity when mixing anti-SARS-CoV-2 vaccines has not been clearly described. In general, several possible mechanisms have been suggested for the higher immune response caused by the mix-and-match strategy. It is suggested that, upon using different vaccine formulations, different arms of the immune system are evoked. Therefore, a combination of cellular and humoral immunity, for instance, can result in higher and prolonged immunity. It has also been seen that higher IgG or neutralizing antibody levels can be achieved using heterologous vaccines, as these vaccines can evoke humoral immunity in different ways [53,54].

Regarding total broadly neutralizing antibodies to SARS-CoV-2 and the type of vaccine, we found lower levels of anti-RDB and anti-S2 serum in patients who received a complete cycle vaccination with Pfizer/BioNTech and the negative anti-N group, but this was not found in the anti-S1 IgG levels.

Recent studies have reported that asymptomatic patients (APs) exhibit a weaker Ab response than patients with severe disease [18,38]. Despite a correlation between anti-RBD and anti-S2 IgG that was reported in [39], asymptomatic vaccinated patients (APs) showed different levels of anti-RBD and anti-S2 IgG anti-SARS-CoV-2 antibodies [14]. Moreover, the sequence of variants may have influenced the serum response to some viral components, favoring subjects with mixed mRNA combinations in negative anti-N subjects.

Researchers encourage the implementation of both antibody and cell-mediated immune response studies in blood samples from vaccinated or unvaccinated individuals who have recovered from an Omicron infection or reinfection [55].

Vaccination strategies that counter immune imprinting are critically needed as a result of the rise of some Omicron variants, specifically XBB and its sub-lineages [56]. The combinations of these mutations could determine further immune-evasion capabilities in those not yet exposed to Omicron. In this regard, urgent action is needed, such as updated vaccine boosters for vaccinated subjects, such as HCWs.

## 5. Conclusions

Our study investigated antibody responses to the RBD, S1, and S2 protein domains in healthy vaccinated populations, such as healthcare workers, considering positive and negative anti-N responses and the types of mRNA vaccines or mixed combinations.

Our results showed a similar response when we analyzed anti-RBD and anti-S1 antibodies in contrast to the anti-S2 profile. We observed that age and anti-S2 IgG were negative and positive significant predictors of anti-N IgG, respectively. This suggests the anti-S2 IgG response generally shows a different dynamic titer in HCW groups with a hybrid immune response.

The correlation between the anti-S2 IgG response and anti-N IgG is probably due to S2′s structural characteristics, as it shows higher protein sequence conservation than RBD and S1.

We found higher levels of total broadly neutralizing antibodies in mixed mRNA combinations. Therefore, the authors consider this finding to be of significant interest in pandemic emergencies when platform vaccinations change according to their local availability.

In addition, the rise of some Omicron variants, specifically XBB and its sub-lineages, encourages a focus on immunity determination in lab-based serology tests to monitor vaccine effectiveness.

## Figures and Tables

**Figure 1 vaccines-11-01136-f001:**
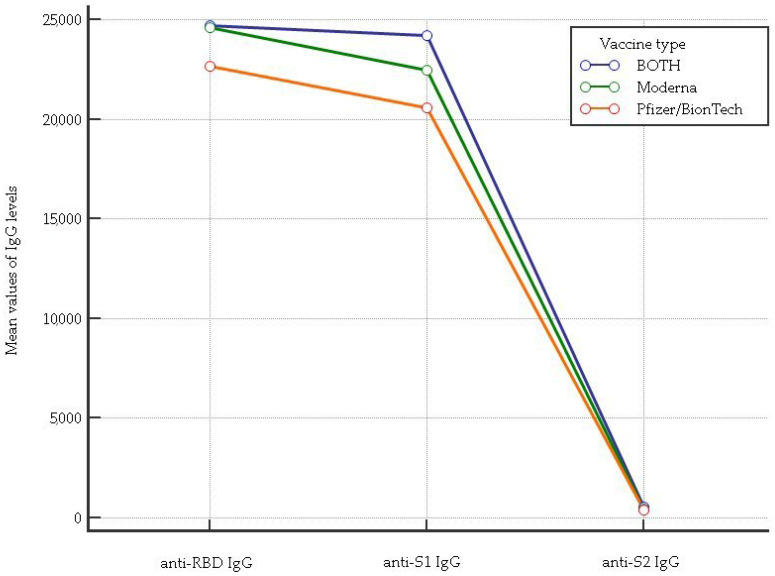
Comparison of anti-RBD IgG, anti-S1 IgG, and anti-S2 IgG levels considering each vaccine type as well as the mixed mRNA vaccine combination.

**Figure 2 vaccines-11-01136-f002:**
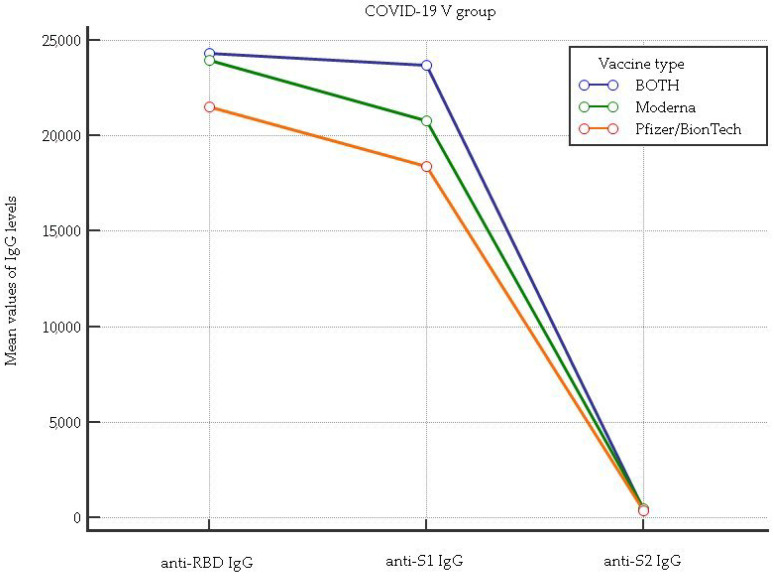
Comparison of anti-RBD IgG, anti-S1 IgG, and anti-S2 IgG levels considering each vaccine type as well as the mixed mRNA vaccine combination in healthcare workers (HCWs) who tested negative for the anti-N (nucleocapsid) protein.

**Figure 3 vaccines-11-01136-f003:**
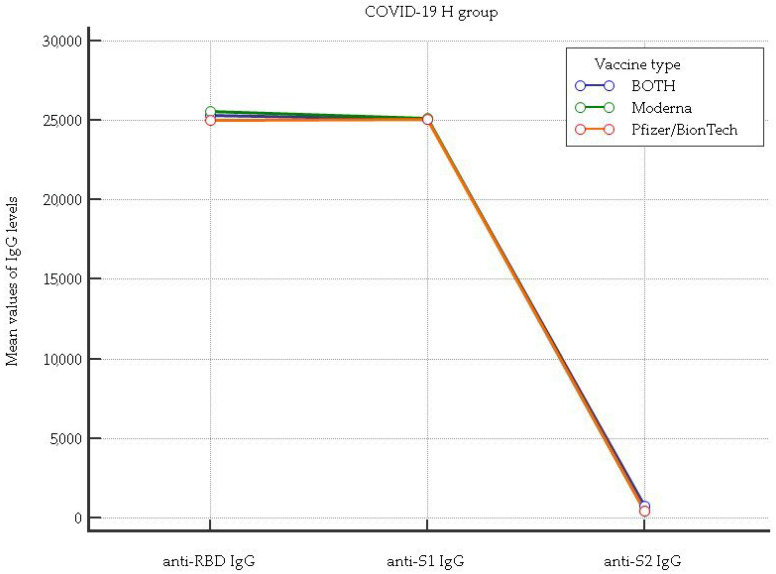
Comparison of anti-RBD IgG, anti-S1 IgG, and anti-S2 IgG levels considering each vaccine type as well as the mixed mRNA vaccine combination in healthcare workers (HCWs) who tested positive for the anti-N (nucleocapsid) protein.

**Table 1 vaccines-11-01136-t001:** General characteristics of 538 vaccinated healthcare workers (HCWs) subjects by age, gender, and detection of antibodies against the receptor-binding domain (RBD), antibody detection against the S protein, S1 domain, S2 domain, and the detection of the level of antibodies against the nucleocapsid (N) protein of SARS-CoV-2.

Parameters	Mean ± SD	Median (IQR)	Titers after Dilution Mean ± SD	Titers after Dilution Median (IQR)	% (N *)
Healthcare workers					538
Age	43.7 ± 12.3	47 (33–53)	-	-	-
Gender					
Male	-	-	-	-	49.4% (266)
Female	-	-	-	-	50.6% (272)
anti-N IgG (U/mL)					
<10	1.7 ± 1.9	0.99 (0.99–0.99)	-	-	65.5% (352)
[10, 100]	34.3 ± 22.8	26 (16–45)	-	-	31.0% (167)
>100	-	-	440.4 ± 338.8	306 (201.75–625)	3.5% (19)
anti-RBD IgG (U/mL)					
<10	5.0 ± 2.8	5 (4, 6)	-	-	0.4% (2)
[10, 100]	49.8 ± 34.2	40 (25.75–76)	-	-	1.1% (6)
>100	-	-	1485.04 ± 311.10	1600 (1600–1600)	98.5% (530)
anti-S1 IgG (U/mL)					
<10	5.0	-	-	-	0.2% (1)
[10, 100]	-	-	-	-	0.0% (0)
>100	-	-	1357.2 ± 434.8	1600 (1284.75–1600)	99.8% (537)
anti-S2 IgG (U/mL)					
<10	5.2 ± 2.4	5(3–7)	-	-	13.5% (73)
[10, 100]	40.0 ± 24.9	33.5(19–54)	-	-	70.3% (378)
>100	-	-	156.8 ± 148.7	125 (80–189)	16.2% (87)

* N = the number and percentage of subjects in parentheses for every antibody-tested level.

**Table 2 vaccines-11-01136-t002:** Characteristics and comparison between the 538 positive and negative healthcare workers (HCWs) to the anti-N (nucleocapsid) protein after the third dose of a COVID-19 vaccination, considering age, gender, anti-RBD IgG, anti-S1 IgG, and anti-S2 IgG.

Parameters	COVID-19 V Anti-N IgG < 10 U/mL	COVID-19 H Anti-N IgG ≥ 10 U/mL	COVID-19 H vs. COVID-19 V *p*-Value (Test)
Healthcare workers (HCWs)	65.4% (352)	34.6% (186)	
Age	45.2 ± 12.2	40.9 ± 12.1	
	48 [38, 55]	43 [30, 51]	0.0001 * (MW)
Gender %Male %Female	52% (183) 48% (169)	44.6% (83) 55.4% (103)	0.10 (C)
anti-RBD IgG (U/mL)	(1N, 351P)	(1N, 185P)	*p* = 1.0 (F)
<10	7.0 ± 0.0 (*n* = 1)	3.0 ± 0.0(*n* = 1)	-
[10, 100]	49.8 ± 34.2 (*n* = 6)	-	-
>100	1600 [1600, 1600] (*n* = 345)	1600 [1600, 1600] (*n* = 185)	*p* < 0.0001 * (MW)
anti-S1 IgG (U/mL)	(0 N, 352 P)	(1N, 185 P)	*p* = 0.35 (F)
<10	-	5.0 ± 0.0(*n* = 1)	-
[10, 100]	-	-	-
>100	1600 [872.5, 1600] (*n* = 352)	1600 [1600, 1600] (*n* = 185)	*p* < 0.0001 * (MW)
anti-S2 IgG (U/mL)	(63 N, 289 P)	(9 N, 176 P)	<0.0001 * (C)
<10	4 [3, 7] (*n* = 63)	6 [4.75, 8.25] (*n* = 9)	*p* = 0.19 (MW)
[10, 100]	28 [17, 47.75] (*n* = 243)	47 [27.25, 64] (*n* = 135)	*p* < 0.0001 * (MW)
>100	114 [82, 213] (*n* = 46)	133 [77.75, 180.5] (*n* = 41)	*p* = 0.93 (MW)

* = significant test, N = negative, P = positive, C = chi-square test, F = Fisher’s exact test, MW = Mann–Whitney test.

**Table 3 vaccines-11-01136-t003:** Logistic regression analysis between the significant parameters in Table 2 with healthcare workers who were anti-N IgG-positive (≥10 U/mL) and anti-N IgG-negative (<10 U/mL).

Logistic Regression	Coefficient	Standard Error	OR	95% CI	*p*-Value
Null model vs. full model					<0.0001 (C)
anti-N IgG/Age	−0.03	0.01	0.97	0.96–0.99	0.0001 *
anti-N IgG/anti-RBD IgG	19.2	11,207.8	>100,000	-	1.0
anti-N IgG/anti-S1 IgG	−38.5	14,133.1	<0.00001	-	1.0
anti-N IgG/anti-S2 IgG	1.5	0.37	4.5	2.2–9.3	0.0001 *
Constant	18.6	8609.8	-	-	1.0

*** = significant test; OR = odds ratio; CI = odds ratio confidence interval at 95%. The null model= −2ln[L_0_], where L_0_ was the likelihood of obtaining the observations if the independent variables did not affect the outcome. The full model= −2ln[L_0_], where L_0_ was the likelihood of obtaining the observations with all independent variables incorporated into the model; C = chi-square test.

**Table 4 vaccines-11-01136-t004:** Comparison among positive responses in the 538 healthcare workers (HCWs) to anti-N IgG, anti-RBD IgG, anti-S1 IgG, and anti-S2 IgG.

Variable	Positive % (n)	Multi-comparison Cochran’s Q Test *p*-Value
(1) anti-N IgG	34.6 (186)	*p* < 0.001 * (Q)
(2) anti-RBD IgG	99.6 (536)	
(3) anti-S1 IgG	99.8 (537)	1 < 2, *p* < 0.05 *, MRD
(4) anti-S2 IgG	86.6 (466)	1 < 3, *p* < 0.05 *, MRD
		1 < 4, *p* < 0.05 *, MRD
		4 < 2, *p* < 0.05 *, MRD
		4 < 3, *p* < 0.05 *, MRD

* = significant test; Q = Cochran’s Q test; MRD = minimum required differences method with Bonferroni *p*-value corrected for multiple comparisons.

**Table 5 vaccines-11-01136-t005:** Comparison among titer dilution values of anti-RBD IgG, anti-S1 IgG, and anti-S2 IgG, stratified according to positive and negative levels of anti-N IgG.

Anti-N IgG	Anti-RBD IgG	Anti-S1 IgG	Anti-S2 IgG	*p*-Value (Test)
*Negative*				*p* < 0.0001 * (KW)
Mean ± SD	1432.6 ± 360.8	1242.3 ± 491.7	178.2 ± 190.9	anti-S2 vs. anti-RBD, *p* < 0.05 * (Co)
Median (IRQ)	1600 [1600, 1600]	1600 [872.5, 1600]	114 [82, 213]	anti-S2 vs. anti-S1, *p* < 0.05* (Co)
Mean rank	433.4	355.2	40.3	anti-S1 vs. anti-RBD, *p* < 0.05 * (Co)
*Positive*				*p* < 0.0001 * (KW)
Mean ± SD	1582.2 ± 141.6	1575.7 ± 127.9	132.9 ± 73.7	
Median (IRQ)	1600 [1600, 1600]	1600 [1600, 1600]	133 [77.75, 180.5]	anti-S2 vs. anti-RBD, *p* < 0.05 * (Co)
Mean rank	226.7	220.2	22.0	anti-S2 vs. anti-S1, *p* < 0.05 * (Co)

* = significant test; KW = Kruskal–Wallis test; Co = Conover post hoc test.

## Data Availability

Not applicable.

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
