# Peer review of "A Serological Analysis of the Humoral Immune Responses of Anti-RBD IgG, Anti-S1 IgG, and Anti-S2 IgG Levels Correlated to Anti-N IgG Positivity and Negativity in Sicilian Healthcare Workers (HCWs) with Third Doses of the mRNA-Based SARS-CoV-2 Vaccine: A Retrospective Cohort Study"

_vaccines, 2023, doi:10.3390/vaccines11071136_

Round 1
Reviewer 1 Report
Nicola Serra and colleagues present a high quality and well-written experimental article focused on serological analysis of the humoral immune response of anti-RBD IgG, anti-S1 IgG, and anti-S2 IgG levels.
Authors retrospectively evaluated the antibody response by multiplex assay in a sample of 538 healthcare workers (HCWs) with documented complete vaccination cycle of three doses of mRNA vaccination and no previous history of infection. Their sample was composed by 49.44% males and 50.56% females, with age included in the range (21-71 yrs) and a mean age of 46.73 yrs. All HCWs sera were collected from April to July 2022 at Sant’Elia Hospital of Caltanissetta to investigate the immunologic response against anti-RBD, anti-S1, anti-S2 and anti-N IgG abs.
Authors found that a significant difference in age between negative and positive HCWs to anti-N IgG. For anti-S2 IgG, a significant difference between negative and positive to anti-N IgG was observed only for positive HCWs with values including in [10(U/mL) -100(U/mL)]; while for anti-RBD IgG and anti-S1 IgG, there was only for positive HCWs with diluted titer. For negative values of anti-N IgG, among titer dilution levels of anti-RBD, anti-S1 and anti-S2 IgG, anti-S2 IgG levels were significantly less than anti-RBD and anti-S1 levels; in addition, anti-S1 IgG levels were significantly less than anti-RBD IgG levels. For anti-N IgG positive levels, only anti-S2 IgG levels were significantly less than anti-RBD IgG and anti-S1 IgG levels. Finally, logistic regression showed that age and anti-S2 IgG were negative and positive predictors of anti-N IgG levels, respectively.
Finally, authors conclude that most anti-N positive HCWs correlated with antibodies against the S2 domain and young subjects. Therefore, the authors suggest including the anti-SARS-CoV-2-S2 in antibody profiles for early identification of asymptomatic infections to reduce the impact of potential SARS-CoV-2 new variants.
Overall, the manuscript is highly valuable for the scientific community and should be accepted for publication.
======================
Other comments to authors:
1) Please check for typos throughout the manuscript.
2) Authors are kindly encouraged to cite the following article that describes novel therapeutic approaches for targeting viral infections, including COVID-19. DOI: 10.3390/biomedicines9010059
Author Response
Dear Reviewer 1, we are grateful for your insightful comments on our paper. We have highlighted all changes within the manuscript.
REVIEWER 1
Comments and Suggestions for Authors
Nicola Serra and colleagues present a high-quality and well-written experimental article focused on the serological analysis of the humoral immune response of anti-RBD IgG, anti-S1 IgG, and anti-S2 IgG levels.
Authors retrospectively evaluated the antibody response by multiplex assay in a sample of 538 healthcare workers (HCWs) with documented complete vaccination cycle of three doses of mRNA vaccination and no previous history of infection. Their sample was composed by 49.44% males and 50.56% females, with age included in the range (21-71 yrs) and a mean age of 46.73 yrs. All HCWs sera were collected from April to July 2022 at Sant’Elia Hospital of Caltanissetta to investigate the immunologic response against anti-RBD, anti-S1, anti-S2 and anti-N IgG abs.
Authors found that a significant difference in age between negative and positive HCWs to anti-N IgG. For anti-S2 IgG, a significant difference between negative and positive to anti-N IgG was observed only for positive HCWs with values including in [10(U/mL) -100(U/mL)]; while for anti-RBD IgG and anti-S1 IgG, there was only for positive HCWs with diluted titer. For negative values of anti-N IgG, among titer dilution levels of anti-RBD, anti-S1 and anti-S2 IgG, anti-S2 IgG levels were significantly less than anti-RBD and anti-S1 levels; in addition, anti-S1 IgG levels were significantly less than anti-RBD IgG levels. For anti-N IgG positive levels, only anti-S2 IgG levels were significantly less than anti-RBD IgG and anti-S1 IgG levels. Finally, logistic regression showed that age and anti-S2 IgG were negative and positive predictors of anti-N IgG levels, respectively.
Finally, authors conclude that most anti-N positive HCWs correlated with antibodies against the S2 domain and young subjects. Therefore, the authors suggest including the anti-SARS-CoV-2-S2 in antibody profiles for early identification of asymptomatic infections to reduce the impact of potential SARS-CoV-2 new variants.
The manuscript is highly valuable to the scientific community and should be accepted for publication.
======================
Other comments to authors:
1) Please check for typos throughout the manuscript.
[REPLY]: Thank you for your suggestion. We checked the manuscript.
2) Authors are kindly encouraged to cite the following article that describes novel therapeutic approaches for targeting viral infections, including COVID-19. DOI: 10.3390/biomedicines9010059
[REPLY]: Thank you for your suggestion. We added the following sentences and references in the discussion section.
Omicron variants have been reported to have potent immune evasion against vaccine-induced neutralizing antibodies and increasing evidence supports the crucial role of the T-cell response to SARS-CoV-2 in controlling the disease. According to previous research reports, the Omicron variant has a strong ability to escape humoral immunity, especially in pts with dysfunction of the human immune response. Recent developments in immunotherapy in viral infections such as Adoptive cell transfer (ACT) with chimeric antigen receptor (CAR)-T cells could represent a prominent example of the treatment of COVID-19.
- Cui, Z., Luo, W., Chen, R., Li, Y., Wang, Z., Liu, Y., Liu, S., Feng, L., Jia, Z., Cheng, R., Tang, J., Huang, W., Zhang, Y., Liu, H., Wang, X., & Li, W. (2023). Comparing T- and B-cell responses to COVID-19 vaccines across varied immune backgrounds. Signal Transduction and Targeted Therapy, 8(1), 1-10. https://doi.org/10.1038/s41392-023-01422-7
- Reeg, D. B., Hofmann, M., Thimme, R., & Luxenburger, H. SARS-CoV-2-Specific T Cell Responses in Immunocompromised Individuals with Cancer, HIV or Solid Organ Transplants. Pathogens, 12(2), 244. https://doi.org/10.3390/pathogens12020244
- Zmievskaya, E., Valiullina, A., Ganeeva, I., Petukhov, A., Rizvanov, A., & Bulatov, E. Application of CAR-T Cell Therapy beyond Oncology: Autoimmune Diseases and Viral Infections. Biomedicines, 9(1), 59. https://doi.org/10.3390/biomedicines9010059

Reviewer 2 Report
The manuscript of Nicola Serra and co-authors is well-written and proposes to include the SARS-CoV-2 subunit 2 in antibody analysis for the identification of asymptomatic COVID-19 infection. The findings of this study would be very useful for new variant-associated COVID-19 waves prevention. There are some comments which can improve the manuscript before it will be published.
1. Authors classified the patient only by sex and age. There is crucial info about the subjects missing: chronic disease history, lifestyle, bad habits such as smoking, alcohol consumption, etc. all of which affect the immune response of vaccination. If the authors don't have that data, they should at least discuss the limitations of the study and how alcohol and smoking affect the COVID-19 vaccination.
2. Are the authors have the information about what exactly vaccines the subjects of the study used (Pfizer/Biotech, or Moderna)? Is any correlation between the type of vaccine and the results?
3. It would be nice to improve the visibility of the manuscript by including the figure with an explanation of how S1, S2, and N-protein act after the booster dose and the first dose of the vaccine.
Author Response
Dear Reviewer 2, we are grateful for your insightful comments on our paper. We have highlighted all changes within the manuscript.
REVIEWER 2
Comments and Suggestions for Authors
The manuscript of Nicola Serra and co-authors is well-written and proposes to include the SARS-CoV-2 subunit 2 in antibody analysis for the identification of asymptomatic COVID-19 infection. The findings of this study would be very useful for new variant-associated COVID-19 waves prevention. There are some comments which can improve the manuscript before it will be published.
- Authors classified the patient only by sex and age. There is crucial info about the subjects missing: chronic disease history, lifestyle, bad habits such as smoking, alcohol consumption, etc. all of which affect the immune response of vaccination. If the authors don't have that data, they should at least discuss the limitations of the study and how alcohol and smoking affect the COVID-19 vaccination.
[REPLY]: Thank you for your suggestion. Thank you for your suggestion. The following sentence and reference were included in the discussion section.
Apart from untreatable factors (age, sex, race), other risk factors which underlie chronic disease, such as improper diet, tobacco consumption, excessive alcohol consumption, insufficient physical activity, sedentary behaviours, and personal or professional stress impact SARS-CoV-2 infection among HCWs and should be analyzed during the COVID-19 pandemic period [49]. In our study, the survey information on SARS-CoV-2 infection included the basic demographic characteristics and only age and sex were analyzed in our statistical database.
- Tani, Y., Takita, M., Kobashi, Y., Wakui, M., Zhao, T., Yamamoto, C., Saito, H., Kawashima, M., Sugiura, S., Nishikawa, Y., Omata, F., Shimazu, Y., Kawamura, T., Sugiyama, A., Nakayama, A., Kaneko, Y., Kodama, T., Kami, M., & Tsubokura, M. Varying Cellular Immune Response against SARS-CoV-2 after the Booster Vaccination: A Cohort Study from Fukushima Vaccination Community Survey, Japan. Vaccines, 11(5), 920. https://doi.org/10.3390/vaccines11050920
- Are the authors have the information about what exactly vaccines the subjects of the study used (Pfizer/Biotech, or Moderna)? Is any correlation between the type of vaccine and the results?
[REPLY]: Thank you for your suggestion. We added comments and figures about the relationships between the vaccine used and anti-RBD IgG, anti-S1 IgG, and anti-S2 IgG antibody levels in different sections of the text as follows and are highlighted in yellow.
section of Abstract:
Results: A significant difference in age between negative and positive HCWs to anti-N IgG was observed. For anti-S2 IgG, a significant difference between negative and positive to anti-N IgG was observed only for positive HCWs with values including in [10(U/mL) -100(U/mL)]; while for anti-RBD IgG and anti-S1 IgG, there was only for positive HCWs with diluted titer. For negative values of anti-N IgG, among titer dilution levels of anti-RBD, anti-S1 and anti-S2 IgG, anti-S2 IgG levels were significantly less than anti-RBD and anti-S1 levels; in addition, anti-S1 IgG levels were significantly less than anti-RBD IgG levels. For anti-N IgG positive levels, only anti-S2 IgG levels were significantly less than anti-RBD IgG and anti-S1 IgG levels. Finally, logistic regression showed that age and anti-S2 IgG were negative and positive predictors of anti-N IgG levels, respectively. The analysis between the vaccine type and mixed mRNA combination showed higher levels of antibodies in mixed vaccinated HCWs. This finding disappeared in the anti-N positive group. Conclusions: Most anti-N positive HCWs correlated with antibodies against the S2 domain and young subjects. Therefore, the authors suggest including the anti-SARS-CoV-2-S2 in antibody profiles can serve as a complementary testing approach to the qRT-PCR for early identification of asymptomatic infections to reduce the impact of potential SARS-CoV-2 new variants. Our serological investigation on the type of mRNA vaccine and mix mRNA vaccines. stress future investigation on serological responses in vaccinated asymptomatic patients exposed or not to previous infection o reinfection for updated vaccine boosters.
Introduction section: This study aimed to investigate antibody levels of spike 1 (S1), spike 2 (S2) and the Receptor Binding Domain (RBD) SARS-CoV-2 antigens in Healthcare Workers with no previous history of infection after six months after receiving their third dose of SARS-CoV-2 mRNA vaccines (Pfizer/Biontech, Moderna) or mixed mRNA vaccines. Moreover, we looked at antibody responses to nucleocapsid to control for possible confounding infection with SARS-CoV-2 and the antibody profile according to the vaccine type and mixed mRNA combination.
Methods section Following the Italian vaccination program, the HCWs received three doses of the mRNA vaccine (Comirnaty (BNT162b2) and Spikevax). In particular, 332 HCWs received Pfizer/BionTech vaccine only, 88 HCWs received Moderna vaccine only, and 44 HCWs received mixed mRNA combination (Pfizer/BionTech and Moderna).
Results section The analysis of levels of anti-RBD IgG, anti-S1 IgG and anti-S2 IgG associated with vaccine type of three doses of mRNA cycle vaccination and mixed mRNA combination, is shown in Figure 1. Regarding the anti-RBD IgG levels, a significant difference among Pfizer/BionTech, Moderna cycle vaccination and mixed mRNA combination was observed (p=0.00014). Particularly, the post hoc Kruskal-Wallis test showed for Pfizer/BionTech cycle vaccination less significant levels of anti-RBD IgG in comparison to Moderna cycle vaccination and mixed mRNA combination (mean rank: 255.1 vs 286.7, p<0.05 and 255.1 vs 297.1, p<0.05). For anti-S1 IgG levels, we found higher levels in HCWs vaccinated with mixed mRNA combination than Pfizer/BionTech and Moderna cycle vaccination (mean rank: 314.1 vs 251.5, p<0.05; 314.1 vs 277.8, p<0.05; respectively). HCWs vaccinated with mixed mRNA combination showed anti-S2 IgG levels greater than single Moderna and Pfizer/BionTech (mean rank: 320.2 vs 291.1, p<0.05; 320.2 vs 245.8, p<0.05; respectively).
In Figures 2 and 3 we showed the anti-RBD IgG, anti-S1 IgG, and anti-S2 IgG levels associated with vaccine type and considering positive and negative healthcare workers (HCWs) to anti-N (nucleocapsid) protein after the third dose of COVID-19 vaccination.
Figure 2 shows for anti-RBD IgG levels a significant difference among Pfizer/BionTech, Moderna and mixed mRNA combination (p=0.00018). Particularly post hoc Kruskal-Wallis test showed a lower level of anti-RBD IgG in Pfizer/BionTech cycle vaccination in comparison to Moderna and mixed mRNA combination (mean rank: 163.8 vs 192.5, p<0.05 and 163.8 vs 203.1, p<0.05). For anti-S1 IgG levels higher levels were found in HCWs with mixed mRNA combination in comparison to Pfizer/BionTech, and Moderna cycle vaccination (mean rank: 223.3 vs 159.7, p<0.05; 223.3 vs 182.1, p<0.05; respectively). For anti-S2 IgG levels, lower levels were found in HCWs vaccinated with Pfizer/BionTech in comparison to Moderna and mixed mRNA combination (mean rank: 160.1 vs 191.2, p<0.05; 160.1 vs 215.7, p<0.05; respectively).
Figure 3 shows for anti-RBD IgG, anti-S1 IgG and anti-S2 IgG levels no significant differences among Pfizer/BionTech, Moderna and mixed mRNA combination (anti-RBD IgG mean rank: 94, 93.3, 93.9, p=0.98; anti-S1 IgG mean rank: 94.3, 94, 91.2, p=0.74, anti-S2 IgG mean rank: 86.6, 98.2, 106.9, p=0.09, respectively).
moreover, Figures 1,2, and 3 were added. Please see the revised version
Discussion section:
During the pandemic waves, the emergence of new variants partially resistant to available vaccines and the report of adverse reactions have forced developing and industrialized countries to start the mix of COVID‐19 vaccines with the hope to immunize a greater percentage of people. Thus, involving also mix mRNA COVID‐19 vaccines. In our study, the analysis between the vaccine type and mixed mRNA combination on anti-RBD IgG, anti-S1 IgG, and anti-S2 IgG levels in all enrolled HCWs confirms that mixed mRNA combination provided greater protection than the use of a single type of vaccine. The authors consider it of greater interest because, in pandemic emergencies, the possibility of performing a vaccination cycle with the same type of vaccine is determined by the local availability of platform vaccination. The underlying mechanism for higher immunity when mixing anti-SARS CoV-2 vaccines has not been clearly described. In general, several possible mechanisms have been suggested for the higher immune response caused by the mix-and-match strategy. It is recommended that by using different vaccine formulations, different arms of the immune system are evoked. Therefore, a combination of cellular and humoral immunity, for instance, can result in higher and prolonged immunity. It has also been seen that higher IgG levels or neutralizing antibodies can be achieved using heterologous vaccines, as these vaccines can evoke humoral immunity in different ways.
Regarding total broadly neutralizing antibodies to SARS-CoV-2 and the type of vaccine, we found lower levels of serum anti-RDB and anti-S2 in patients with complete cycle vaccination with Pfizer/BioNTech Pfizer and in the anti-N negative group, but these findings didn’t find in anti-S1 IgG levels.
Recent studies have reported that asymptomatic patients (AP) exhibit a weaker Ab response than patients with severe disease [18,39]. Despite a correlation between anti-RBD and anti-S2 IgG reported in AP [39] asymptomatic vaccinated patients (AP) showed different levels of anti-RBD and anti-S2 IgG anti-SARS-CoV-2 antibodies [38]. Moreover, the sequence of variants could be influenced the serum response to some viral components, favoring subjects with mixed mRNA combinations in anti-N negative subjects.
Researchers encourage the implementation of both antibody and cell-mediated immune response studies in blood samples from vaccinated or unvaccinated individuals who had recovered from an Omicron infection or reinfection [54].
Vaccination strategies that counter immune imprinting are critically needed due to the rise of some Omicron variants, specifically XBB and its sublineages [55]. The combinations of these mutations could determine further immune-evasion capability in those not yet exposed to Omicron. Action needs, such as updated vaccine boosters, to be taken urgently in this regard, also in vaccinated subjects such as HCWs.
- Conclusions
Our study investigated the antibody response to RBD, S1 and S2 protein domains in healthy vaccinated populations such as healthcare workers, considering positive and negative anti-N responses and types of mRNA vaccines or mixed combinations.
Our results showed a similar response when we analyzed anti-RBD and anti-S1 antibodies in contrast to the anti-S2 profile. We observed that age and anti-S2 IgG were negative and positive significant predictors of anti-N IgG, respectively. This suggests the anti-S2 IgG response generally offers a different dynamic titer in HCWs groups with the hybrid immune response.
The correlation between anti-S2 IgG response and anti-N IgG is probably due to structural S2 characteristics, showing higher protein sequence conservation levels than RBD and S1.
We found higher levels of total broadly neutralizing antibodies in mixed mRNA combinations. Therefore, the authors consider this finding of significant interest in pandemic emergencies when platform vaccination changes according to local availability.
In addition, the rise of some Omicron variants, specifically XBB and its sublineages, encourages a focus on immune determining in lab-based serology tests to monitor vaccines’ effectiveness.
- It would be nice to improve the visibility of the manuscript by including the figure with an explanation of how S1, S2, and N-protein act after the booster dose and the first dose of the vaccine.
[REPLY]: Thank you for your question. this study was developed on patients at the end of the vaccination cycle, i.e. after the third dose of the vaccine. Unfortunately, we have no data on the intermediate steps

Reviewer 3 Report
Although the COVID-19 has been effectively controlled, the study relating to breakthrough infection is of significance to the development of advanced vaccine in the future. The manuscript entitled "Serological analysis of the humoral immune response of anti-RBD IgG, anti-S1 IgG, and anti-S2 IgG levels correlated to anti-N IgG positive and negative in Sicilian Healthcare Workers (HCWs) with third doses of mRNA-based SARS-CoV-2 vaccine: a retrospective cohort study" describe an interesting finding of the antibody levels against S1, S2, RBD and N in healthcare workers with no previously reported infection. And they found that Most anti-N positive HCWs correlated with antibodies against the S2 domain and young subjects. The authors undertook an interesting study. However, the data are not well analyzed and the discussion could be further improved. This interesting study can be published in form of short communication or letters after revision.
Other comments:
1. The manuscript requires major proof reading by a native speaker.
2. The main limitation of the manuscript is the lack of data on serum neutralizing antibody titers and their association with the titers of antibodies against S1, S2, RBD, especially anti-N antibody. These correlation will give more valuable information for the interpretation of breakthrough infection of SARS-CoV-2.
3. As currently detecting nucleic acid or viral antigen is more convenient and powerful method for SARS-CoV-2, it should be careful to make suggestions, such as including the anti-SARS-CoV-2-S2 in antibody profiles for early indemnification of asymptomatic infections.
4. In addition, the manuscript is somewhat verbose, some analysis of the data, such as the relationship between age and the level of anti-N antibody, has little practical significance. It is widely known that the elderly are more susceptible to COVID-19.
5. Some rows in the tables are not full aligned.
Author Response
Dear Reviewer 3, we are grateful for your insightful comments on our paper. We have highlighted all changes within the manuscript.
REVIEWER 3
Comments and Suggestions for Authors
Although COVID-19 has been effectively controlled, the study relating to breakthrough infection is of significance to the development of advanced vaccines in the future. The manuscript entitled "Serological analysis of the humoral immune response of anti-RBD IgG, anti-S1 IgG, and anti-S2 IgG levels correlated to anti-N IgG positive and negative in Sicilian Healthcare Workers (HCWs) with third doses of mRNA-based SARS-CoV-2 vaccine: a retrospective cohort study" describes an interesting finding of the antibody levels against S1, S2, RBD and N in healthcare workers with no previously reported infection. And they found that most anti-N positive HCWs correlated with antibodies against the S2 domain and young subjects. The authors undertook an interesting study. However, the data are not well analyzed and the discussion could be further improved. This interesting study can be published in the form of short communication or letters after revision.
Other comments:
- The manuscript requires major proofreading by a native speaker.
[REPLY]: Thank you for your suggestion. We used MDPI's editing service.
- The main limitation of the manuscript is the lack of data on serum-neutralizing antibody titers and their association with the titers of antibodies against S1, S2, and RBD, especially anti-N antibodies. These correlations will give more valuable information for the interpretation of breakthrough infection of SARS-CoV-2.
[REPLY]: Thank you for your question. The entire manuscript is based on the search for correlations between anti-N IgG, anti-RBD IgG, anti-S1 IgG, and anti-S2 IgG. In fact, apart from table 1 which is purely descriptive, we showed in table 2 the possible relationship between anti-N IgG, and age, gender, anti-RBD IgG, anti-S1 IgG, and anti-S2 IgG. In table 3 we performed a multivariate analysis with a logistic regression between anti-N IgG positive (≥10 U/mL) and negative (<10 U/mL) and the significant parameters of table 2. In table 4 an additional analysis was performed to individualize the differences among anti-N IgG, anti-RBD IgG, anti-S1 IgG, and anti-S2 IgG on the positive response of 538 healthcare workers. Finally, in table 5 we performed a comparison among titers dilution values of anti-RBD IgG, anti-S1 IgG and anti-S2 IgG, stratified according to positive and negative levels of anti-N IgG, justifying the need for a separate analysis only on titers dilution values. Finally, we added in this revised version the comments and figures about the relationships between the type of vaccines and mixed mRNA vaccination and antibodies levels.
- As currently detecting nucleic acid or viral antigen is a more convenient and powerful method for SARS-CoV-2, it should be careful to make suggestions, such as including the anti-SARS-CoV-2-S2 in antibody profiles for early identification of asymptomatic infections.
[REPLY]: thanks for your suggestion. the following sentence in the abstract section was modified.
Therefore, the authors suggest including the anti-SARS-CoV-2-S2 in antibody profiles can serve as a complementary testing approach to the qRT-PCR for early identification of asymptomatic infections to reduce the impact of potential SARS-CoV-2 new variants.
- In addition, the manuscript is somewhat verbose, and some analysis of the data, such as the relationship between age and the level of anti-N antibody, has little practical significance. It is widely known that elderly patients are more susceptible to COVID-19.
[REPLY]: Thank you for your question. The manuscript has been reviewed by a native speaker. The relationship between age and the level of anti-N antibody was added for the completeness of our manuscript; but as you will see our manuscript shows many additional results and analyses. Moreover, as the population of Italian healthcare workers is older than in other countries, this can also contribute to understanding the influence of the age factor. According to the suggestion of previous reviewers, the discussion section was revised as follows and is highlighted in yellow (Please in the attach file).
- Some rows in the tables are not fully aligned.
[REPLY]: Thank you for your suggestion. We formatted them.

Round 2
Reviewer 2 Report
The manuscript was improved and could be accepted for publication